# Epigenetics Approaches toward Precision Medicine for Idiopathic Pulmonary Fibrosis: Focus on DNA Methylation

**DOI:** 10.3390/biomedicines11041047

**Published:** 2023-03-28

**Authors:** Wiwin Is Effendi, Tatsuya Nagano

**Affiliations:** 1Department of Pulmonology and Respiratory Medicine, Faculty of Medicine, Universitas Airlangga (UNAIR), Surabaya 60132, Indonesia; 2Department of Pulmonology and Respiratory Medicine, Universitas Airlangga Teaching Hospital, Surabaya 60015, Indonesia; 3Pulmonology and Respiratory Medicine of UNAIR (PaRU) Research Center, Universitas Airlangga Teaching Hospital, Surabaya 60015, Indonesia; 4Division of Respiratory Medicine, Department of Internal Medicine, Kobe University Graduate School of Medicine, 7-5-1 Kusunoki-cho, Chuo-ku, Kobe 650-0017, Japan

**Keywords:** epigenetics, epigenomics, DNA methylation, long noncoding, microRNA, histone modification, idiopathic pulmonary fibrosis, chronic pulmonary diseases

## Abstract

Genetic information is not transmitted solely by DNA but by the epigenetics process. Epigenetics describes molecular missing link pathways that could bridge the gap between the genetic background and environmental risk factors that contribute to the pathogenesis of pulmonary fibrosis. Specific epigenetic patterns, especially DNA methylation, histone modifications, long non-coding, and microRNA (miRNAs), affect the endophenotypes underlying the development of idiopathic pulmonary fibrosis (IPF). Among all the epigenetic marks, DNA methylation modifications have been the most widely studied in IPF. This review summarizes the current knowledge concerning DNA methylation changes in pulmonary fibrosis and demonstrates a promising novel epigenetics-based precision medicine.

## 1. Introduction

IPF is a chronic, devastating, and irreversible lung disease that is characterized by microinjury-induced alveolar epithelial cell stress, progressive pathogenic myofibroblast differentiation, imbalanced macrophage polarization, and the extensive deposition of the extracellular matrix (ECM) [1,2,3]. The progression of patients with IPF is associated with lung function decline, progressive respiratory failure, high mortality, recurrent acute exacerbations, and an overall poor prognosis [4,5,6]. The morphological hallmark of IPF on histopathological and/or radiological is usual interstitial pneumonia (UIP), composed of heterogeneous areas of normal-appearing lung intermixed with collagenized fibrosis in sub-pleural and paraseptal, a honeycombing pattern, and ECM-producing myofibroblasts termed fibroblast foci (FF) [7,8].

A recent hypothesis stated that recurrent injuries drive the aberrant activation of epithelial cells to transdifferentiate into fibroblast epithelial-mesenchymal transition (EMT), which might induce fibrosis independently of inflammatory events [9,10]. Even though there is no implicit mechanism, several shreds of evidence emphasize that alveolar epithelial injury induced by environmental triggers results in lung fibrosis. Recurrent microenvironment injury on senescent epithelial cells in genetically susceptible individuals leads to the aberrant activation of fibroblasts, accumulating ECM, and fibrosis [11].

The pathogenic mechanisms involved in the initiation, development, and progression of IPF are unclear. However, many studies have demonstrated that dynamic interactions of genetic susceptibility, environmental factors, and host risk factors in older individuals contribute to epigenetic pro-fibrotic reprogramming, resulting in the development of IPF [12]. Hey et al. found a strong association between the microenvironment-driven epigenetic changes that could induce macrophage inflammation and polarization [13].

Omics-based approaches, including high-throughput technologies that provide snapshots of a holistic view of the molecules that make up a cell, tissue, or organism, consist of (1) genomics, measuring deoxyribonucleic acid (DNA) sequence variation; (2) epigenomics, focusing on the genome-wide characterization of reversible modifications of DNA or DNA alterations; (3) transcriptomics, evaluating the standard of ribonucleic acid (RNA) expression; (4) proteomics, determining protein expression or its chemical changes; (5) metabolomics, assessing metabolite/small molecule levels; and (6) microbiomics, investigating all the microorganisms of a given community [14,15]. Pulmonary disease omics studies mainly focus on tissue- and cell-specific omics data and have identified several fundamental mechanisms that underlie pulmonary biological processes, disease endotypes, and appropriate novel therapeutics for selected individuals [16].

Epigenomics is a technique for analyzing gene expression through epigenetic mechanisms, including DNA methylation, RNA, and histone modification [17]. These components interact and stabilize each other; therefore, the disruption of epigenetic nucleosomes can lead to their inappropriate expression, resulting in epigenetic disorders [18]. Epigenetics and epigenomics help explain how our environment affects our phenotype. Epigenomics studies commonly use methods such as Hi-C, a comprehensive technique developed to capture chromosome conformation, and another tool for whole genome methylation profiling: MBD-isolated genome sequencing (MiGS) [19]. Essential technical and experimental parameters that should be considered when designing epigenomic experiments are reviewed in detail by these authors [20].

Precision or personalized medicine is designed for patients who do not respond to conventional therapy due to genetic heterogeneity and epigenetic alterations [21]. The research-based precision medicine approach identifies the complex genetic, molecular, environmental, and behavioral variables that could provide greater efficacy and tolerability in IPF patients [22]. This review provides a brief outlook on identifying the epigenetic marks of IPF via epigenome-wide association studies that might also inform precision medicine approaches.

## 2. Epigenetics

Genetics focuses on heritable changes in gene activity or function due to the direct alteration of the DNA sequence. In contrast, epigenetic mechanisms, such as DNA methylation, histone modifications, or miRNA expression, regulate heritable gene activity or function changes that can be transferred without modifying the DNA sequence itself [23,24]. Epigenetics provides beneficial information concerning the adaption of genes to environmental changes or other stresses [25]. Furthermore, epigenetic modifications can be transferred from generation to generation, providing an alternative mechanism for disease inheritance and risk [26,27].

Whereas epigenetics refer to how and when single genes or sets are turned on and turned off, epigenomics analyzes epigenetic changes associated with genome-wide profiles and effects [28]. Several factors and processes affect the imbalance of epigenetic mechanisms, including development in utero and during a lifetime, environmental factors, aging, and lifestyle [29,30]. In comparison with genetic changes, epigenetic changes are acquired in a gradual rather than a short process [31].

Epigenetics is responsible for numerous cellular functions, such as the regulation of gene expression and transcription, cell growth and differentiation, and chromosome remodeling and inactivation [32,33,34]. Many studies demonstrated an epigenetic link between environmental stimuli and gene expression as an adaptation of genes in response to ecological changes without modifying the DNA sequence [35]. However, those processes become unbalanced in many diseases, such as cancer and fibrosis. Aberrant epigenetic regulations have been reportedly associated with the progression of chronic respiratory diseases, such as chronic obstructive pulmonary disease (COPD) and lung fibrosis [36,37,38]. Many epigenomics studies in IPF patients reported aberrant epigenetic influences, including DNA methylation, miRNA, and histone modifications, which mediate alterations to the DNA without impacting the genomic sequence [38,39,40]. Our review systematically provides current knowledge concerning various epigenetic aberrations in IPF, particularly DNA methylation. However, DNA methylation interacts with histone modifications and miRNA to activate or silence gene expression, contributing to disease development and progression.

### 2.1. DNA Methylation

The mammalian DNA methylome pattern is a dynamic process of inheritable pre-transcriptional modifications that are balanced by two antagonizing processes, including methylation and demethylation [41]. DNA methylation is typically reconfigured after fertilization during zygote formation and gametogenesis and then re-established during embryogenesis [42]. DNA methylation is essential for normal development and several vital processes, including genomic imprinting, X-chromosome inactivation, the suppression of repetitive element transcription, transposition, and preservation of chromosome stability [43,44,45]. Therefore, DNA methylation is the most intensively investigated epigenetic element that influences gene activities. The environment can affect physiological or pathological changes in DNA methylation, resulting in modified global and gene-specific DNA methylation [46]. DNA methylation regulates chromosomal and extrachromosomal DNA functions in response to environmental exposures [47].

DNA methylation is a reversible process that refers to transferring the methyl (CH_3_) group from S-adenosyl methionine (SAM) to the fifth carbon of a cytosine of the CpG dinucleotide forming 5-methylcytosine (5mC) [48] (Figure 1). DNA methylation occurs at cytosines in any context of the genome; however, most of the genome does not contain CpG sites. Cytosine methylation in mammalian cells occurs predominantly in CpG dinucleotide clusters, which are called CpG islands, the regions enriched in CpGs that coincide with gene promoters [49]. DNA methylation is also found at non-CpG sites or non-CpG methylation, including CpA, CpT, and CpC [50]. DNA methylation is catalyzed by a family of DNA methyltransferases (DNMTs), including DNMT1, DNMT2, DNMT3a, DNMT3b, and DNMT3L, which are responsible for setting up and maintaining methylation patterns at specific genome regions [51]. Furthermore, DNA methylation is also controlled by methyl-binding proteins (MBPs), such as the methyl-CpG-binding domain (MBD) proteins family, Kaiso and Kaiso-like proteins, and the SRA domain proteins that interact with MBD proteins [52,53].

DNA demethylation can occur actively or passively in mammals. Active DNA demethylation means the direct removal of a methyl group from cytosines through the oxidation of 5mC by ten-eleven translocation enzymes (TETs) to form 5-hydroxymethylcytosine (5hmC), 5-formylcytosine (5fC), and 5-carboxylcytosine (5caC), followed by the excision of 5fC and 5caC by thymine DNA glycosylase (TDG) coupled with base excision repair [54,55] (Figure 1). This alteration can also take place passively by a reduction or inhibition activity of DNMTs during DNA replication [56,57]. Demethylation is linked to transcription factor binding and proper gene expression [58]. According to their functions, enzymes that establish, recognize, and remove DNA methylation, such as DNMTs, MBPs, and TET, could be referred to as writers, readers, and erasers, respectively [23,59]. These epigenetic writers catalyze the process and introduce various chemical modifications to the DNA and histones; readers may identify, recognize, and bind to methyl groups to influence gene expression and eraser, modify, and remove the methyl group [23,60]. DNA methylation regulates pro- or anti-fibrotic gene expression in organ fibrosis [59]. Hypomethylated gene promoters correspond to increased gene expression, while hypermethylated refers to decreased gene expression [61].

#### 2.1.1. DNMTs

DNA methylation patterns are established through de novo methylation by DNMT3a, DNMT3b, and the maintenance of methylated cytosine via DNMT1 [62]. However, many partners of DNMTs are involved in both the regulation of DNA methylation and DNMT recruitment, such as the DNMT3L/DNMT3a complex, which is related to de novo DNA methylation and maintains global DNA methylation that is regulated by the DNMT1/PCNA/UHRF1 complex [63].

DNMT1 preferentially methylates hemimethylated DNA and is responsible for copying DNA methylation patterns from the mother to the daughter strands during the S phase or DNA replication [64]. The role of DNMT2 in DNA methylation remains unclear, though it appears to have significance in the methylation of tRNA [65]. In contrast to DNMT1, DNMT3a and DNMT3b prefer unmethylated CpG dinucleotides, which perform de novo methylation during embryogenesis, and set up genomic imprints during germ cell development [66]. DNMT3L acts as a cofactor and stimulates de novo methylation by DNMT3A or DNMT3b [67]. It should be noted that the maintenance vs. de novo function of these enzymes is not absolute; maintenance and de novo methylation usually cooperate to maintain a stable methylation pattern [68].

DNMTs have exhibited their essential roles in multiple organ fibrosis. The DNMT1-mediated suppression of cytokine signaling 3 (SOCS3) leads to the deregulation of STAT3 to promote cardiac fibroblast activation and collagen deposition in diabetic cardiac fibrosis [69]. The down-regulation of SOCS3 promotes the activation of STAT3-related fibroblast-to-myofibroblast transition. The DNMT3A-induced silencing of SOCS3 expression stimulates TGF-β-dependent fibroblast activation in experimental systemic sclerosis (SSc) murine models [70]. A recent study found that the peroxisome proliferator-activated receptor-γ (PPARγ) expression was low, but increased DNMT 1/DNMT3a and PPARγ promoter hypermethylation in IPF patients; therefore, the inhibition of DNA methylation that restored PPARγ led to attenuated pulmonary fibrosis [71]. The aberrant expression of DNMTs patterns is associated with pulmonary fibrosis, although the exact mechanisms underlying this pathogenesis remain elusive.

#### 2.1.2. MBPs

DNA methylation has an essential regulatory function to suppress non-transcribed genes stably in differentiated adult somatic cells. The detailed mechanism of DNA methylation that represses gene expression remains unknown. The repressive effects of DNA methylation on the expression of genes involve two mechanisms, directly by DNA methylation that prevents transcriptional regulators and indirectly via MBP-binding methylated DNA and the recruitment of co-repressors, such as histone deacetylases (HDAC), to silence the genes [59]. MBPs recognize DNA methylation and translate the DNA methylation signal into appropriate functional states, such as chromatin organization and transcriptional repression [53]. Three families of the MBPs protein are involved in gene repression or the inhibition and binding of the transcription factor to DNA [23]. MBPs not only bind specifically to CpG dinucleotides but also participate in the translation of DNA methylation and histone modifications [72].

The most widely studied MBPs are MBD protein families, including MeCP2, MBD1-MBD6, STEDB1, STEDB2, BAZ2A, and BAZ2B [73]. MeCP2 is a well-characterized MBD protein that has made progress in defining its biochemical properties. MeCP2 controls myofibroblast transdifferentiation and inhibits fibrosis. MeCP2 inhibits the α-tubulin acetylation-related fibroblast proliferation in cardiac fibroblasts through HDAC6 [74]. MeCP2 expression was increased in macrophages; therefore, MeCP2 expression was inhibited through siRNA-inhibited M2 macrophage polarization [75]. MeCP2 appears to mediate pro-fibrotic and anti-fibrotic effects, although more studies are needed to define its role in the pathogenesis of pulmonary fibrosis.

#### 2.1.3. TETs

TET enzymes TET 1, TET 2, and TET 3 mediate active DNA demethylation mechanisms. Although the role of TET in organ fibrosis is not clear, many studies have reported associations between DNA demethylation, TET enzyme activity, and fibrosis. TET3 and TGF-β are pro-fibrotic factors. The inhibition of TET3 signaling attenuates hepatic fibrosis by abrogating the positive feedback loop TET3/TGF-β1 to promote pro-fibrotic gene expression [76]. On the contrary, other studies have proven the anti-fibrotic role of TET in kidney fibrosis. Hypoxia triggers renal fibrosis due to the reduced expression of TET and induced Klotho methylation; therefore, the administration of a low-dose sodium hydrosulfide (NaHS) restores TET activity [77].

The suppression of specific genes through aberrant promoter methylation contributes to organ fibrosis. TET-mediated demethylation inhibits fibrosis. Xu et al. showed the gene-specific demethylation of four gene promoters by TET3CD using high-fidelity Cas9 (dHFCas9)-TET3CD, which attenuates kidney fibrosis [78]. Excessive TET-dependent GDF7 (growth differentiation factor 7) hypomethylation up-regulated pro-fibrotic genes through the bone morphogenetic protein receptor type 2 (BMPR2)/Smad signaling pathway, α-smooth muscle actin (α-SMA), and fibronectin (FN) [79]. Furthermore, a recent student demonstrated that silencing anti-fibrotic factors, and the RAS protein activator-like 1 (RASAL1), depends on the R-loop Gm15749 formation that interacts with the demethylation complex consisting of GADD45g and TET3 [80].

## 3. DNA Methylation in Fibrosis

Extensive alterations in DNA methylation profiles are known to be involved in the pathogenesis of pulmonary fibrosis. DNA methylation microarray demonstrated a higher DNA methyltransferase expression in lung tissue samples of IPF patients [81]. IPF fibroblasts exhibited significant heterogeneity in global DNA methylation patterns compared to non-fibrotic control cells [82]. Additionally, a study identified extensive alterations in gene expression-associated DNA methylation that were involved in fibroproliferation [83]. Recently, a genome-wide DNA methylation study found that CpGs methylation was responsible for cell adhesion, molecule binding, chemical homeostasis, surfactant homeostasis, and receptor binding categories in the pathogenesis of IPF [84].

Global methylation patterns in IPF are very different from that of the control samples and significantly overlap with methylation changes observed in lung adenocarcinoma samples [85]. DNA methylation biomarkers also overlap between IPF, other ILD, cancer, and COPD; hence, it is unlikely that molecular discrimination between these diseases can be achieved using a single marker [86]. Interestingly, low-methylation lung squamous cell carcinoma (SCC) significantly correlated with IPF to show a bad outcome [87]. Cigarette smoking-associated aberrant methylation contributed to pulmonary fibrosis [88].

Many cell types are well known to be involved in the process of pulmonary fibrosis. The essential cells driving the development and progression of pulmonary fibrosis are epithelial, fibroblasts and myofibroblasts, and alveolar macrophages. Epigenetic regulation has a variety of ways. This study focuses on DNA methylation regulation in fibroblasts, epithelial, and macrophages.

### 3.1. DNA Methylation and EMT

EMT is a cellular and molecular process in which epithelial cells lose their epithelial identity (apical-basal polarity and adhesion), which is characterized by down-regulated epithelial markers, including E-cadherin, occludin, and claudin-1. In contrast, fibroblast-specific genes, such as α-SMA, N-cadherin, fibroblast-specific protein 1 (FSP-1), and type I collagen, are up-regulated [10,89]. A hallmark of EMT is the repression of E-cadherin: a transmembrane glycoprotein encoded by the epithelial marker gene CDH1. DNA methylation might be involved in regulating EMT. During EMT induction, EMT-transcriptional factors (EMT-TFs), including ZEB1, SNAIL, and TWIST, interact with DNMTs and undergo the hypermethylation of CpG islands in the CDH1 promoter [90,91]. The DNA hypermethylation of the CDH1 through DNMTs was correlated with tumor progression [90,92]. In addition, the knockdown of DNMTs was equated with the transfection of siDNMT1, DNMT3a, or DNMT3b, which reversed the TGF-β1-induced suppression of E-cadherin expression and the induction of α-SMA, vimentin, and fibronectin expression [93].

In normal lung epithelial wound healing, this process ends with the apoptosis of myofibroblasts and inflammation reduction. Yet, aberrant responses to tissue injury may turn into lung fibrosis. Bidirectional EMT cross-talk assists the pro-fibrogenic positive feedback loop, while epithelial cells become “vulnerable and sensitive to apoptosis” and myofibroblasts become “apoptosis-resistant and immortal”, resulting in fibrosis progression instead of wound resolution [94]. DNA methylation mediated the down-regulation of genes involved in apoptosis. The hypermethylation of the Caspase 8 promoter by DNMT1 and DNMT3b was associated with apoptosis resistance in cancer cells [95,96]. Cisneros et al. showed the silencing expression of pro-apoptotic p14ARF in the fibroblast of the patient with IPF due to hypermethylation [97]. Along with these changes, DNA methylation was linked with the dysregulation of apoptosis and EMT, leading to the development of lung fibrosis processes (Figure 2).

### 3.2. DNA Methylation and Myofibroblast Differentiation

Under normal physiology, in response to epithelial injury, the cell releases various pro-inflammation and fibrotic cytokines that are responsible for local inflammation and the activation of fibroblasts. Fibroblasts are mesenchymal cells that are recruited to and accumulate at the injured site and undergo a change in phenotype to highly proliferative and contractile myofibroblasts.

Lung myofibroblasts are heterogeneous in terms of their origins. The predominant sources of myofibroblasts are resident fibroblasts (lipofibroblasts, matrix fibroblasts, and alveolar niche cells) and pericytes (residing within basal membranes or perivascular linings), along with minor sources such as hematopoietic CXCR4+ fibrocytes, alveolar epithelial cells (AECs), endothelial cells (ECs), and mesenchymal stem cells (MSCs) [98]. Under the influence of environment-specific factors, cytokines, such as TGF-β, fibroblast-to-myofibroblast transition/transdifferentiation (FMT) changes lung resident fibroblast behavior/phenotypes to another type of normal differentiated cell and the downstream effects at the tissue level [99]. Myofibroblasts, characterized by the expression of contractile α-SMA, regulate connective tissue remodeling by producing and modifying ECM components, such as fibronectin and collagen [100]. Furthermore, one of the mechanisms that may account for FMT is DNA methylation.

DNA methylation modulates myofibroblast differentiation (Figure 3). Several genes and signaling pathways, such as those involving ErbB, focal adhesion, and MAPK, underlie mechanisms for DNA methylation alteration in myofibroblast differentiation and influence the synthesis of ECM [101]. DNA methylation regulates gene expression to facilitate the formation of fibroblastic foci and lung fibrosis [47]. DNMT is an essential regulator of α-SMA gene expression during myofibroblast differentiation. The specific effect of DNA methylation in regulating the α-SMA gene is unknown. During the development of cardiac fibrosis, He et al. found that TGF-β could inhibit DNMT1-mediated DNA methylation expression, leading to the overexpression of α-SMA [102]. TGF-β1 also regulates lung fibroblast differentiation through the hypermethylation of the “fibrosis suppressor” gene, Thymocyte differentiation antigen-1 (*THY-1*); therefore, DNMT1-attenuated TGF1-mediated THY-1-inducing α-SMA fiber formation is silenced [103]. In addition, the expression of α-SMA was significantly increased by adding IL-6 [104]. A recent study revealed that DNMTl knockdown suppressed the DNA methylation-related myofibroblast phenotype via inhibiting an IL-6/STAT3/NF-κβ positive feedback loop [105].

MBD protein families have been suggested to play an essential role in myofibroblast differentiation. The hypermethylation of the transcriptional regulator, c8orf4, decreased the capacity of fibrotic lung fibroblasts to up-regulate COX-2 expression and COX-2-derived PGE2 synthesis [106]. Evidence showed that MeCP2 binds with the I*κ*B*α* promoter and induces the α-SMA promoter [107,108]. The inhibition of MeCP2-associated TGF-β1 significantly blocks the expression levels of α-SMA and fibronectin production during myofibroblast differentiation [109]. In addition, suppressing MeCP2 or MBD2 might reduce pro-fibrotic factors. Wang et al. determined higher MBD2 levels in different types of pulmonary fibrosis patients, including COVID-19, systemic sclerosis-associated interstitial lung disease, and IPF. Therefore, deleting the MBD2 gene attenuated bleomycin-induced lung injury and fibrosis [110]. MBD2 promoted fibroblast differentiation via repressing the expression of the erythroid differentiation regulator (Erdr) 1 [111].

Interestingly, other studies showed the anti-fibrotic effects of MBDs protein families. The in vitro study of diffuse cutaneous scleroderma (SSc) showed that the overexpression of MeCP2 via the modulating genes PLAU, NID2, and ADA suppressed fibroblast proliferation, migration, and myofibroblast differentiation [112]. Perhaps the final effects of DNA methylation on pulmonary fibrosis depends on the hypomethylation of pro-fibrosis genes and hypermethylation of anti-fibrosis genes [111]. This evidence revealed that the role of the epigenetic gene transcription regulator in myofibroblast differentiation was complex.

Hypoxia can alter DNA methylation patterns and contribute to the fibrotic process. Global DNA methylation was detected in hypoxic fibroblasts and was associated with increased DNMT1 and DNMT3A expression and decreased THY-1 expression [113,114]. Hypoxia altered DNA methylation, which is characterized by the increased expression of TGF-β1 and DNMT1, and significantly decreased RASAL1 [115]. Chronic hypoxia induces oxidative stress. Recently, the administration of the antioxidant enzyme extracellular superoxide dismutase (EC-SOD) has been shown to alleviate the hypoxia-induced DNA methylation of the Ras association domain family 1 isoform A (RASSF1A) through the Ras/ERK pathway [116].

### 3.3. DNA Methylation and Macrophage Polarization

Pulmonary macrophages consist of monocyte-derived alveolar (AMs) and lung tissue-resident alveolar or interstitial macrophages (IMs). Both lung tissue-resident macrophages and monocyte-derived macrophages can be polarized into classically activated macrophages (M1) or alternatively activated macrophages (M2) by their capacity to induce inflammatory or anti-inflammatory immune responses, respectively [117]. The heterogeneity of macrophages regulates the development of pulmonary fibrosis from the early phases of injury and the fibrotic phase. The M2 macrophage, instead of the M1 phenotype, is involved in the progression of lung fibrosis [118]. After the resolution of the lung injury, monocyte-derived AM persisted and expressed higher pro-inflammatory and pro-fibrotic functions, and therefore, leading to the depletion of monocyte-derived AM ameliorates lung fibrosis [119].

DNA methylation alterations in IPF lungs have been well documented, but the role of DNA methylation in macrophages needs to be better defined. Many studies have shown that DNA methylation plays a fundamental role in macrophages, including the regulation of injury-induced inflammation and the control of fibroblast activation in the wound-healing response. Chen et al. examined DNA methylation changes in lung macrophages and found that changes in DNA methylation patterns drove the dysregulation of innate immune cells in cystic fibrosis [120]. Profiling DNA methylation from healthy subjects and IPF patients showed that aberrant macrophage polarization played a crucial role in developing IPF, but epigenetic alterations were not associated with accelerated aging [38]. DNMTs are associated with the activation of gene expression during macrophage phenotypic changes. During the development of IPF, DNMTs are associated with inappropriate gene expression activation, resulting in the opposite roles of M1/M2 polarization.

DNMTs can act as pro- and anti-fibrotic by regulating M2 polarization. Yang et al. revealed that DNMT3a and DNMT3b suppress the Proline-Serine-Threonine Phosphatase Interacting Protein 2 (PSTPIP-2) by hypermethylation results, which caused a mixed induction of hepatic macrophage M1 and M2 [121]. In obesity, DNMT3b regulates M2 macrophage polarization, and the deletion of DNMT3b induces M2 macrophage polarization [122]. On the contrary, another study provided different results. Qin et al. found that DNMT3b ameliorates the development of bleomycin-induced pulmonary fibrosis by limiting M2 macrophage polarization [123]. In this case, DNMT3b acts as a negative regulator of M2 macrophage polarization. The role of DNMT3-associated macrophage polarization in organ fibrosis might have different functions. DNMT3 limits M2 macrophage polarization and alleviates the progression of lung fibrosis, yet DNMT3 induces M2 polarization and enhances liver fibrogenesis.

Consistent with the involvement of DNMTs in macrophage phenotype change, MBPs interact with methylated DNA to regulate the expression of multiple genes. However, MeCP2 can accelerate both M1 and M2 polarization. MeCP2 might tend to M1 in the early phase of inflammation. The deletion of MeCP2 down-regulated M1 differentiation and inhibited the expression of iNOS and the secretion of cytokines that are pro-inflammatory in acute lung injury (ALI) [124]. MeCP2 regulates the gene expression of macrophages; thus, MeCP2 deficiency leads to the dysregulation of macrophage polarization [125]. MeCP2 accelerates M2 polarization by inhibiting Ship expression and enhancing the phosphatidylinositol 3-kinases/protein kinase B (PI3K/PKB) signaling pathway in bleomycin-induced pulmonary fibrosis [110]. Furthermore, silencing MeCP2 using siRNA blunted M2 macrophage polarization to elevate IRF4 expression [75]. Conversely, MeCP2 directly or indirectly promoted the differentiation of M0 macrophages to M1 or M2 macrophages and the polarization of M2 to M1 macrophages [126]. Therefore, we might suggest that MeCP2 stimulates M2 macrophages to promote the progression of lung fibrosis but protect against renal fibrosis via M1 polarization.

Little is known about the impact of TET proteins on macrophage polarization. TET regulates macrophage differentiation and M1-polarization [127]. Therefore, the inhibition of the TET protein using dimethyloxallyl glycine (DMOG) promotes M2 polarization by up-regulating Arg1, Fizz1, and Ym1 [128].

## 4. Targeted Gene and Personalized Medicine

IPF is an epigenetic disease in which environment-associated epigenomic alterations may lead to aberrant regulation in fibroblasts, epithelial, and macrophages during lung wound healing. The growing knowledge of epigenetic mechanisms should facilitate precision medicine that is closer than ever to patients with IPF. Precision medicine means personalized medicine. This concept emphasizes the customization of medical practice, including prevention and treatment strategies, focusing on the individual with regard to omics-based biomarkers and individual and environmental factors [129,130]. There have been attempts to apply personalized medicine to manage pulmonary diseases, such as asthma, COPD, lung cancer, cystic fibrosis, and ILD [131,132,133].

Personalized medicine revolutionizes patient care as targeted genes can be associated with a particular phenotype or disease. Precision medicine focuses on the use of the ‘omics’ approach to direct more personalized diagnosis through the use of biomarkers and cost-effective strategies based on personal clinical and molecular information [134,135]. From the perspective of targeting epigenetics in precision medicine, the strategy for epigenetics therapy encompasses enhancing, silencing, and repressing gene expression [136].

DNA methylation is the most crucial of all the epigenetic mechanisms, since it controls gene expression. The pathogenesis of IPF is complex and interconnected. Therefore, targeting several genes involved in the DNA methylation-associated pathogenic mechanisms of IPF, including inflammation, epithelial apoptosis, myofibroblast differentiation, macrophage polarization, and extracellular matrix metabolism, is very crucial [137,138]. Several specific genes exhibit the DNA hypermethylation alteration that is involved in the development of IPF and their potential future for targeted therapeutics. Manipulating the epigenetic control of gene expression related to fibrosis could be a promising therapeutic target for IPF.

*THY-1* is a suppressor gene that maintains cell-stromal balance in normal lung fibroblasts. Fibroblasts from normal lungs are THY-1 positive, while THY-1 expression in IPF fibroblasts is negative or minimal. THY-1 overexpression reduced pulmonary fibrosis and down-regulated fibroblast markers such as MMP-2, occludin, α–SMA, and vimentin [139]. DNA hypermethylation down-regulates THY-1 expression. TGF-β1 is a potential modifier of the DNA methylome; therefore, stimulation with the TGF-β1 modified DNA methylation of normal and IPF fibroblasts are characterized by the significantly decreased expression of THY-1 and the up-regulation of DNMT1, DNMT3a, and DNMT3b [140]. Hypermethylation in the promoter of THY-1 suppressed the *THY-1* gene and enhanced the anti-apoptotic roles of lung fibroblasts, resulting in ECM deposition and lung scar formation [141]. In addition, hypoxia-induced global DNA hypermethylation decreased the expression of THY-1 [114].

The PGE2 function inhibits the activation of the fibroblast. However, there are not many fibroblasts in IPF and they are resistant to PGE2. The diminished capacity of IPF and SSc lung fibroblasts to the up-regulation of COX-2-derived PGE2 synthesis results from hypermethylation and the silencing of the transcriptional regulator, c8orf4 [106]. PGE2 increases global DNA methylation via DNA methyltransferase; therefore, the inhibition of COX-2-derived PGE2 and DNMT inhibits cell growth [142].

Various studies have shown that the alteration of DNA methylation is associated with fibrogenic gene expression. Aberrant DNA methylation patterns, hyper- and hypo-methylation, are involved in the pathogenesis of pulmonary fibrosis. While DNA hypermethylation contributes to repressing anti-fibrotic genes, hypomethylation results in the induction of pro-fibrotic genes. Therefore, epigenetic-modifying drugs are attractive therapeutic agents in IPF for reversing DNA methylation. 5-aza-2′-deoxycytidine (5aza), a DNA methylation inhibitor, displayed beneficial effects in organ fibrosis and cancers. 5aza inhibited the profibrotic factor TGF-β1-induced hypermethylation and repression of erythropoietin in the pericytes of kidney fibrosis [143]. In addition, 5aza reduced global DNA methylation in chemo-sensitive cancer cells [143]. Targeting DNMT 1/DNMT3a and the peroxisome proliferator-activated receptor-γ (PPAR-γ) axis with 5-aza can demethylate the PPAR-γ promoter, restore PPAR-γ loss, and alleviate fibrotic lung [71].

5aza is a potent inducer of DNA de-methylation and has been approved by FDA for epigenetics-therapeutics in some cancer. However, their potential effect of inhibiting fibrosis in IPF still needs to be fully understood. The newest evidence suggests that hypermethylation is not always associated with transcriptional repression, but is also related to high transcriptional activity [144].

## 5. Conclusions

Environmental circumstances and genetics influence epigenetic mechanisms. Epigenetic changes via DNA methylation, miRNA, and histone modification induce gene-phenotype alterations. Extensive alterations in DNA methylation profiles are known to be involved in the pathogenesis of IPF.

Aberrant DNA methylation is associated with various cells and mechanisms that underlie the pathogenesis of IPF. Hypermethylation mediated by DNMTs and MBPs resulted in silencing of the expression of the anti-fibrotic gene. These alterations lead to the induction of myofibroblast differentiation, apoptosis resistance, EMT, and the secretion of macrophage-associated pro-fibrotic factors, promoting the progression of lung fibrosis. Activating a pro-fibrotic phenotype provides a positive feedback loop. In addition, hypoxia-related epigenetic alteration plays an essential role in shaping fibrosis. Indeed, the epigenetic gene transcription regulator in myofibroblast differentiation is complex; therefore, the final effects of DNA methylation on pulmonary fibrosis depend on the hypomethylation of pro-fibrosis genes and hypermethylation of anti-fibrosis genes.

DNA methylation is reversible; thus, finding mechanisms or drugs with which to control the methylation pattern is crucial. DNMT inhibitors or other modifiers that modulate DNA methylation are currently available only for the preclinical stage. Further studies to understand the alteration and methylation pattern involved in the pathogenesis of IPF are beneficial in developing novel targeted/personalized treatments focused on DNMTs, MBPs, or TET to prevent or even reverse the fibrotic process.

## Figures and Tables

**Figure 1 biomedicines-11-01047-f001:**
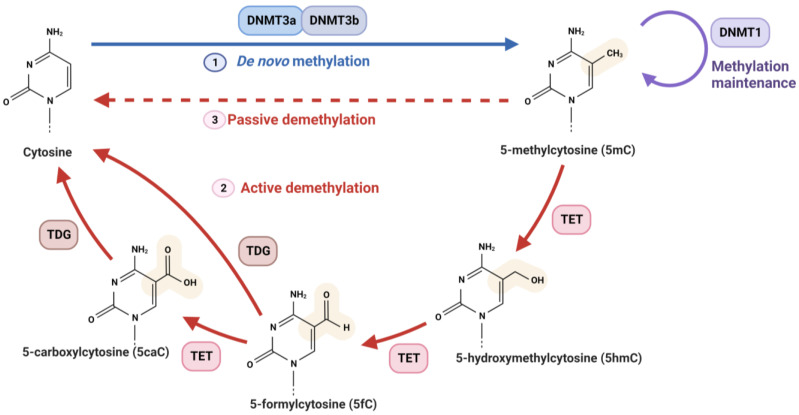
DNA methylation-demethylation. (1) Methylation pathways. S-adenyl methionine (SAM) donor methyl (CH3) group to the fifth carbon of cytosine residue forming 5-methylcytosine (5mC) catalyzed by DNA methyltransferases (DNMTs) family. DNMT3a and DNMT3b perform de novo methylation during embryogenesis, while DNMT1 maintains a DNA methylation pattern during replication. (2) Active demethylation is the direct removal of a methyl group from cytosines through the oxidation of 5mC by ten-eleven translocation enzymes (TETs) to form 5-hydroxymethylcytosine (5hmC), 5-formylcytosine (5fC), and 5-carboxylcytosine (5caC) followed by the excision of 5fC and 5caC by thymine DNA glycosylase (TDG). (3) Passive demethylation corresponds to the reduction or inhibition activity of DNMTs during DNA replication.

**Figure 2 biomedicines-11-01047-f002:**
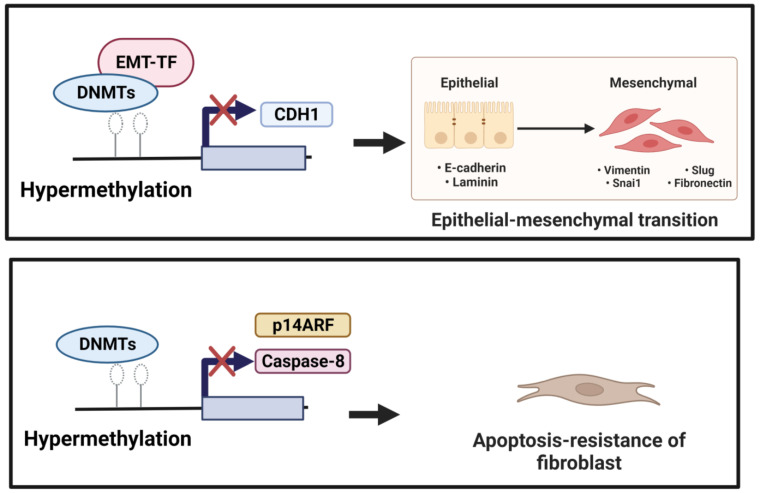
(1) Induction of EMT. Down-regulation of CDH1 (anti-fibrotic factor) is mediated by EMT-TF, which recruits DNMTs to the CDH1 promoter performing focal hypermethylation of the CpG islands in the CDH1 promoter, leading to decreased gene expression of CDH1. (2) Reduction in apoptotic activity. Hypermethylation mediated by DNMTs of the pro-apoptotic factors, including Caspase 8 and p14 ARF, could underlie resistance to fibroblast apoptosis.

**Figure 3 biomedicines-11-01047-f003:**
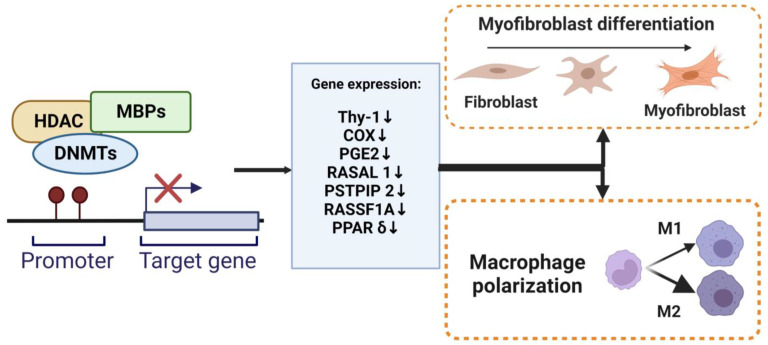
DNA methylation regulate myofibroblasts and macrophages. The methylation of promoter DNA can repress gene expression by MBPs, recruiting specific macromolecular complexes that contain histone deacetylases (HDAC), DNMTs, and other transcriptional co-repressors (Co-Rep). Several genes that function as an anti-fibrotic and are associated with myofibroblast differentiation and macrophage M2 polarization have been shown to undergo silencing due to DNA hypermethylation.

## Data Availability

Data available in a publicly accessible repository.

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
