# Peer review of "Epigenetics Approaches toward Precision Medicine for Idiopathic Pulmonary Fibrosis: Focus on DNA Methylation"

_biomedicines, 2023, doi:10.3390/biomedicines11041047_

Round 1

Reviewer 1 Report

The review is generally well-written and informative. One minor comment is that the basics of epigenetics is already very well established in the literature and there is no need to cover that again in section 2. This section should be shortened extensively, so that the content will be more focused on the subject of this review. 

Author Response

Review report (reviewer 1)

The review is generally well-written and informative. One minor comment is that the basics of epigenetics is already very well established in the literature and there is no need to cover that again in section 2. This section should be shortened extensively, so that the content will be more focused on the subject of this review. 

Author`s response

Thank you for your appreciation. We have already corrected several minor issues listed by reviewer 1 by

Reviewer 2 Report

The review summarizes the role of DNA methylation in idiophatic lung fibrosis. The topic is interesting and contributes to epigenetic knowledge of this pathology. The review is well written and organized, including most of bibliography that exist about the field.  

Major points 

The review is focused in DNA methylation. The description of microRNAs and histone modifications is too long. This section should be only a short introduction about the different epigenetic mechanisms. Please reduce this section. 

Minor points 

Review the numbers of subsections in 2. Epigenetics because there are some subsections labelled as 2.1 and others as 3.1.1 or 3.1.2. 

Author Response

Review report (reviewer 2)

The review summarizes the role of DNA methylation in idiophatic lung fibrosis. The topic is interesting and contributes to epigenetic knowledge of this pathology. The review is well written and organized, including most of bibliography that exist about the field.  

Major points 

The review is focused in DNA methylation. The description of microRNAs and histone modifications is too long. This section should be only a short introduction about the different epigenetic mechanisms. Please reduce this section. 

Minor points 

Review the numbers of subsections in 2. Epigenetics because there are some subsections labelled as 2.1 and others as 3.1.1 or 3.1.2. 

Author`s response

Thank you for your appreciation. We have already corrected several minor and major issues listed by reviewer 2

Major issues

We extensively reduced the miRNA and histone modification discussion and focused on DNA methylation alteration-related IPF.

Minor issues

Errors in the subsection are already corrected.

Reviewer 3 Report

Dear editor,

The review is focused on the role of DNA methylation and other epigenetic mechanisms in idiopathic pulmonary fibrosis.

The review is well written and summarizes the data on the topic. The main issue is a discordance between the title “Epigenetics Approaches toward Precision Medicine…” and the content. At chapter 4, Targeted gene and personalized medicine, it is not clear how the personalized medicine for idiopathic pulmonary fibrosis may be applied. The general ways and methods of gene therapy and target medicine for lungs should at least be mentioned. How to change a particular gene activity in a particular organ, lung? Clearly, the proposed usage of 5aza or epigenetic modifying genes on a level of a whole organism is not a good idea as it will damage normal processes and may lead to cancer. Basically, just four paragraphs are directly related to the title.

Overall, the data presented in the review is interesting. I’d recommend the article for publication after the revision and English language proofreading.

Minor remarks:

There are some spelling mistakes that should be corrected. The authors should check throughout the text whether they are talking about genes or proteins and mark the genes by italics. In several cases the data is definitely about genes/RNA expression levels but there are no italics.

L107-108 The sentence is too general and not clear. Zygote formation, gametogenesis and embryogenesis are very far away in the terms of time, and follow in different order. Also, the authors should put a reference to the particular case of DNA methylation, imprinted genes, and that takes part during gametogenesis.

L200 The serial comma should be put after “BAZ2A” for consistency. Same applies to other places in the text.

L220 “4” should be written as “four” as there is no two-digit numbers in the sentence.

L230, L257 Please check whether miRNA at the beginning of the sentence should be “miRNA” or “MiRNA”.

L336 The word “than” seems to be grammatically incorrect.

L369 The correct form seems to be “involving”.

L377 “hypermethylation the “fibrosis suppressor” gen” should be “hypermethylation of the “fibrosis suppressor” gene”.

L424 “instead M1 phenotype” should be “instead of M1 phenotype.

L469 should be “regulates”.

L479 “Thy-1” – human genes/proteins should be in uppercase.

Author Response

Review report (reviewer 3)

Dear editor,

The review is focused on the role of DNA methylation and other epigenetic mechanisms in idiopathic pulmonary fibrosis.

The review is well written and summarizes the data on the topic. The main issue is a discordance between the title “Epigenetics Approaches toward Precision Medicine…” and the content. At chapter 4, Targeted gene and personalized medicine, it is not clear how the personalized medicine for idiopathic pulmonary fibrosis may be applied. The general ways and methods of gene therapy and target medicine for lungs should at least be mentioned. How to change a particular gene activity in a particular organ, lung? Clearly, the proposed usage of 5aza or epigenetic modifying genes on a level of a whole organism is not a good idea as it will damage normal processes and may lead to cancer. Basically, just four paragraphs are directly related to the title.

Overall, the data presented in the review is interesting. I’d recommend the article for publication after the revision and English language proofreading.

Minor remarks:

There are some spelling mistakes that should be corrected. The authors should check throughout the text whether they are talking about genes or proteins and mark the genes by italics. In several cases the data is definitely about genes/RNA expression levels but there are no italics.

L107-108 The sentence is too general and not clear. Zygote formation, gametogenesis and embryogenesis are very far away in the terms of time, and follow in different order. Also, the authors should put a reference to the particular case of DNA methylation, imprinted genes, and that takes part during gametogenesis.

L200 The serial comma should be put after “BAZ2A” for consistency. Same applies to other places in the text.

L220 “4” should be written as “four” as there is no two-digit numbers in the sentence.

L230, L257 Please check whether miRNA at the beginning of the sentence should be “miRNA” or “MiRNA”.

L336 The word “than” seems to be grammatically incorrect.

L369 The correct form seems to be “involving”.

L377 “hypermethylation the “fibrosis suppressor” gen” should be “hypermethylation of the “fibrosis suppressor” gene”.

L424 “instead M1 phenotype” should be “instead of M1 phenotype.

L469 should be “regulates”.

L479 “Thy-1” – human genes/proteins should be in uppercase.

Author`s response

Thank you for your appreciation. We have already corrected several minor issues listed by reviewer 1 (using the track changes).

Minor revisions

For the role of DNA methylation in zygote formation, gametogenesis, and embryogenesis, the authors decide to remove the paragraph to minimize misunderstanding.

Major revisions

To revise the discordance between the title and the content, the authors add new information and basic principles about precision medicine and targeting several genes involved in the DNA methylation-associated pathogenic mechanisms of IPF.
